# Effect of Feed Supplementation with Tripotassium Citrate or Sodium Chloride on the Development of Urinary Calcium Oxalate Crystals in Fattening Pigs

**DOI:** 10.3390/vetsci9110614

**Published:** 2022-11-06

**Authors:** Joris Vrielinck, Geert P. J. Janssens, Ilias Chantziaras, An Cools, Dominiek Maes

**Affiliations:** 1Faculty of Veterinary Medicine, Ghent University, 9000 Ghent, Belgium; 2Veterinary Practice, Hospitaalstraat 38, 8906 Ieper-Elverdinge, Belgium

**Keywords:** pigs, urolithiasis, urinalysis, feed, control

## Abstract

**Simple Summary:**

The study examined whether dietary supplements such as citrate or NaCl have an influence on the development of urinary calcium oxalate crystals in fattening pigs. There were three examination groups: a control group, a citrate group and a NaCl group. Parameters of comparison were performance (average daily gain and feed intake, feed conversion ratio, water consumption), blood parameters (bone resorption and bone formation markers, calcium, phosphorus, potassium and parathyroid hormone), macroscopic (stones, grit) and microscopic (crystals) examination of the urine and biochemical urinalysis of samples taken at the farm and at the slaughterhouse. There were no beneficial effects of feed supplementation with citrate or NaCl on the development of crystals in the urine of fattening pigs. On the contrary, citrate has a clear promoting influence on the development of alkaline calcite crystals. Substantial differences are found between microscopic and biochemical urinalysis results from samples taken at the farm and at the slaughterhouse. Slaughterhouse samples do not fully reflect the situation on the farm.

**Abstract:**

The present study investigated whether dietary supplementation of tripotassium citrate or NaCl reduced the prevalence of calcium oxalate dihydrate (COD) uroliths in fattening pigs on a farm with a high prevalence of COD uroliths. Each group (control, TPC, NaCl) consisted of three batches of approximately 260 fattening pigs each. Performance, water intake, markers for bone resorption (CTX) and bone formation (osteocalcin) and urinalysis from samples taken at the farm and in the slaughterhouse were investigated. Performance parameters, feed and water intake, CTX and osteocalcin were not significantly different between the groups (*p* > 0.05). The main crystals found were struvite, COD, calcite and amorphous crystals. The prevalence of COD crystals was lower in samples from the slaughterhouse in each group. Microscopic and biochemical examination of urine showed large differences between samples from the farm and the slaughterhouse. In conclusion, there were no beneficial effects of feed supplementation with TPC or NaCl on the prevalence of COD crystals but TPC has a clear promoting influence on the development of alkaline calcite crystals. Urinalysis from samples taken at the slaughterhouse does not fully reflect the situation on the farm.

## 1. Introduction

Urolithiasis is defined as the presence of calculi or uroliths in the urinary passages [1]. In contrast to other domestic animals, urolithiasis is not frequently detected in pigs of all ages. The condition may damage the epithelia of the urinary tract and, in severe cases, lead to obstruction of urinary passages [2]. Previous research has shown that urolithiasis is present in a large proportion of male fattening pigs [3]. Macroscopic and microscopic abnormalities were detected in 8.4% and 52.8% of the samples, respectively. Magnesium ammonium phosphate (struvite), calcium oxalate dihydrate (COD), calcium carbonate (calcite), calcium oxalate monohydrate (COM) and amorphous crystals were detected. Analysis of stones showed the presence of COD in all samples in different proportions. The calcium content of examined stones was always considerable (up to 34%).

The pathogenesis of urolithiasis in pigs is not yet fully elucidated. Factors known to predispose to the formation of uroliths include diet, urinary pH, reduced water intake, urinary stasis and preexisting urinary tract diseases. Different nutritional factors can either promote or inhibit the development of uroliths or microscopic crystals in the urine [4,5,6]. Changes in diet composition are, therefore, a very useful tool to control or prevent urolithiasis in pig farms. However, almost no information is available on pigs. The present study focuses on the addition of tripotassium citrate (TPC) and sodium chloride (NaCl) to the pig diet as possible control measures against COD urolithiasis. In humans, TPC given orally acts as an inhibitor for the development of COD crystals in the urine. Higher concentrations of citrate in the urine will bind calcium to form soluble calcium citrate so that less Ca is available to bind oxalate or phosphate [7,8]. In addition, TPC is metabolised to bicarbonate and leads to the alkalinisation of the blood and urine [8,9]. The higher urine pH decreases the risk for COD crystals, which are more easily formed in acid urine. However, more alkaline urine might increase the risk for the formation of crystals such as calcite and calcium phosphate.

Sodium chloride (NaCl) is frequently supplemented in pig feed to enhance feed intake [10]. Salt intake stimulates thirst and enhances water intake in humans, resulting in more diluted urine [11,12]. This is considered important in the prevention of urolithiasis. Calcium follows sodium in its tubular resorption or excretion by the kidney and, therefore, NaCl supplementation might lead to calciuria and calcium crystal formation [13,14]. A higher salt intake might, therefore, also increase the risk for urolithiasis, although the dilution effect may dominate and outweigh the effects of this pathway. Dietary salt loading is also associated with hypocitraturia, an inhibitor for calcium crystal formation [15]. Given the many physiological similarities between humans and pigs, it can be expected that similar processes might also take place in pigs.

Calcium in uroliths and urinary crystals can originate from bone tissue (endogenous calcium) and/or from the diet (exogenous calcium). This can be investigated through the determination of the circulating concentrations of bone resorption markers such as beta-isomerised carboxyl-terminal telopeptide cleaved from type 1 collagen during bone resorption (CTX) and bone formation marker such as osteocalcin, a non-collagen bone protein produced by osteoblasts during bone formation [16,17,18].

The present study investigated whether dietary supplementation of NaCl and TPC reduced the prevalence of COD uroliths in fattening pigs on a pig farm with a high prevalence of oxalate urolithiasis. We also assessed whether NaCl and TPC supplementation influenced bone mineralisation, given their influence on urinary calcium and pH [13,19].

## 2. Material and Methods

### 2.1. Study Farm

A semi-closed farm with 170 Topigs-20 sows and 750 finishing pigs (sows and barrows, Topigs-20 x Belgian Piétrain) was selected from the 50 farms that were included in a previous study [3]. In this farm, COD crystals had been detected in 50% of the examined urinary samples of slaughter pigs. There were three identical compartments for fattening pigs on the farm, each with a capacity of 264 pigs. There was a fully slatted floor and the stocking density was 0.75 m^2^ per pig.

During the fattening period, five different mash feeds were given according to the weight of the pigs: feed 1 (25–40 kg), feed 2 (40–55 kg), feed 3 (55–70 kg), feed 4 (70–95 kg) and feed 5 (95 kg until slaughter). The feeds were delivered by the same feed mill company throughout the study. The composition of the different mash feeds is shown in Appendix A (Appendix A).

### 2.2. Experimental Design

To control urolithiasis on the farm, an on-farm intervention study was conducted during the fattening period (from approximately 12 weeks of age until slaughter) by comparing the normal situation on the farm (control feed) with supplementation of the feed with either NaCl or TPC. The inclusion levels of TPC and NaCl were 2.5 kg/ton (0.25% supplementation) and 2.0 kg/ton (0.2% supplementation), respectively. Based on the literature [10,20] and internal discussion between the authors, these levels were expected to confer a beneficial effect without inducing possible negative effects. The inclusion level of either NaCl or TPC remained the same for the five feeds that were used throughout the entire fattening period in the respective treatment groups.

In total, nine successive batches of approximately 255–260 fattening pigs each (24 pens with 11–12 pigs per pen) were included. They were raised over a period of 15 months, namely from June 2019 until September 2020. There were three batches (replicates) of each treatment, and the treatments were randomly distributed over the three compartments. Hence, each replicate of each treatment group was raised in a different compartment (Table 1).

Each compartment was filled with pigs at once, and the pigs were delivered to the slaughterhouse two times (each time half of the group), with an interval of 14 days. To determine our sample size, we used historic prevalence data from the farm. We assumed that 50% of the pigs would show urolithiasis in the control group and determined that 150 animals per group would be sufficient to achieve a power of 80% for detecting a relative risk of 0.68 between the treatment groups and the control group (50% versus 34% prevalence) at a 5% level of significance (two-sided test).

The mineral composition of the feed provided during the last three weeks of the fattening period was analysed from each replicate of the three treatment groups by means of induction-coupled plasma spectrometry. The kation–anion balance (Na+K-Cl) and the dietary electrolyte balance (dEB: Na+K-Cl+Ca+Mg-S-P) are expressed in meq/kg and determined by dividing the values of the ions in mg/kg by the corresponding atomic mass and valence.

There was one feeding trough per pen, and one drinking nipple (flow rate 1–1.5 L/min) was located in each feeding trough. The pigs had ad libitum access to the feed and to the drinking water. The drinking water originated from the public supply, and the quality of the drinking water was examined once during the study.

### 2.3. Parameters of Comparison

#### 2.3.1. Performance and Water Consumption

At the beginning and at the end of the fattening period, all pigs of the nine batches were weighed at compartment level. The average daily gain (ADG) (kg/pig/day) during the fattening period was calculated by subtracting the starting weights from the final weights (including dead pigs), divided by the number of days lived by the group during the fattening period.

The feed intake was measured for each compartment separately. Each feed supply during the entire fattening period was recorded, allowing us to calculate the average daily feed intake. The feed conversion ratio (FCR) was calculated by dividing the feed intake by the difference between final (including dead pigs) and starting body weights. A corrected FCR was also calculated for a standardised fattening trajectory of 25 to 115 kg.

The number of dead pigs was recorded, with the date of death as well as the estimated weight of the dead animals. Data from dead pigs were used to calculate the mortality rate.

Water consumption was recorded for each batch on a weekly basis during the entire fattening period, allowing us to calculate the average water use per pig per day.

#### 2.3.2. Blood Parameters

At the age of 24 weeks, which is approximately three weeks prior to slaughter, 10 pigs were randomly selected and blood samples (clotted blood) were taken by puncture of the jugular vein. The serum was collected from the blood samples and stored at −20 °C until analysis. The samples were taken by the herd veterinarian in connection with routine health monitoring on the farm, and, therefore, no approval of the ethical committee for animal experiments was needed.

The following parameters were analysed in the serum: calcium, potassium, phosphorus, parathyroid hormone (PTH), the bone resorption marker CTX (cross-linked C-telopeptide of type I collagen) and the bone formation marker osteocalcin. PTH and CTX were examined by the ECLIA method (Electro Chemiluminescence Immuno Assay) and osteocalcin by a radio immune assay (RIA). Calcium, potassium and phosphorus were determined by spectrophotometry. All analyses were performed in the Laboratory for Clinical Biology at the university hospital of Ghent University, Belgium, except for the osteocalcin analysis which was done at the Royal Health Center (GD) at Deventer, The Netherlands.

#### 2.3.3. Urinalysis of Samples Taken at the Farm

Approximately two weeks before the slaughter of the fattening pigs, five to ten pigs from each replicate were selected by convenience sampling and urine samples from sows and barrows were taken. To this end, a two-metre-long stick with at the end a plastic drinking beaker was developed, allowing the collection of spontaneously voided urine by the animals. Upon collection, the urine was stored at 4 °C and transported to the veterinary practice for macroscopic and microscopic examination and urinalysis.

Urinalysis was performed on fresh urine within 1–2 h after sampling, as cooling for 24 h can promote the formation of various types of crystals that are not representative of the in vivo situation [21,22].

The urine was evaluated macroscopically for the presence of (1) stones, (2) grit and (3) white flakes.

The urine was centrifuged for 45 s at 9800 rotations per minute (Statspin VT rotor, VetQuip, Sydney Australia). The sediment was examined microscopically under two magnifications, namely 100× and 400× for the presence of urinary crystals such as COD, COM, calcite and amorphous crystals. The abundance of calcite crystals was assessed using a scoring system, with scores ranging from one to five. Score 1 was used when 1 or 2 crystals were present per microscopic field, score 2 for 3–5 crystals, score 3 for 6–20 crystals, score 4 for 21–100 crystals and score 5 for cases when more than 100 crystals were present per microscopic field.

The urine samples were also examined biochemically for Ca, P, Mg, Na, Cl, K, citrate and creatinine in the Laboratory for Clinical Biology at the university hospital of Ghent University, Belgium. Creatinine was determined by an enzymatic method, and citrate by an enzymatic colourimetric method. The values of Ca, P, Mg, Na, Cl and K were determined by photometric methods and citrate was expressed as standardised to creatinine levels, (expressed as mmol/L per gram creatinine/L). Urinary pH and specific gravidity (SG) were determined on all samples at the veterinary practice by using urinary dipsticks (Combur 10 test strips, Roche Diagnostics GmbH, Mannheim, Germany) and a handheld refractometer (Analogue refractometer type ORA SPB, Kern GmbH, Balingen, Germany), respectively.

#### 2.3.4. Urinalysis of Samples Taken at the Slaughterhouse

From every batch of pigs, approximately 40 urinary bladders were collected at the slaughterhouse according to the protocol described in our previous prevalence study [3]. After urine collection, the bladder was opened, and the presence of (1) stones, (2) grit and (3) white flakes in the bladder content were assessed. The microscopic examination of the sediment and the urinalysis were performed in the same way as the urine samples taken on the farm. The urinalysis was not done on all samples collected at the slaughterhouse, but on a subsample.

### 2.4. Statistical Analysis

All continuous data were checked for normality and homogeneity of variance of the selected model by examining histograms and normal probability plots of residuals. Normally distributed data were analysed with linear regression models, using batch and treatment (control, feed supplemented with NaCl, supplemented with TPC) as a fixed effect. Bonferroni correction was applied for pairwise comparisons. When equal variances could not be assumed, appropriate post hoc tests were performed to indicate differences between the three groups (Tamhane T2 test). Data on weekly drinking water intake were analysed using linear mixed models with repeated measurements (autoregressive repeated covariance type), treatment was included as fixed and batch as a random effect. All other data were analysed with the nonparametric (Kruskal–Wallis) test.

Generalised linear (binomial logit) models were used for all two-sided tests of equality for column proportions. Fixed variables included in each of the models were the treatment group and farm. Sequential Bonferroni testing was used to correct pairwise comparisons. All analyses were performed in SPSS 27©, Armonk, New York.

## 3. Results

### 3.1. Mineral Composition of Feed and Drinking Water Quality

As expected, the sodium and chloride content was highest in the NaCl group, while the potassium content was highest in the TPC group (*p*-values between 0.05 and 0.10) (Table 2). The mean Na+K-Cl balance and mean dEB value, estimates for the alkalinity of the feed, were numerically highest in the control group (control 146 and 218; TPC 135 and 203; NaCl 131 and 191, respectively) (Table 2) (*p* > 0.05).

The bacteriological and biochemical parameters of the drinking water were within the normal ranges (Appendix A).

### 3.2. Performance Parameters and Water Consumption

There were no significant differences in performance parameters (*p* > 0.05) between the three treatment groups. Additionally, the feed and water intake and the water-to-feed ratios were similar between the three groups (Table 3).

### 3.3. Blood Parameters

The results of the parameters measured in the serum of the pigs from the three treatment groups (control, TPC, NaCl) are shown in Table 4.

The differences between the groups were not statistically significant (*p* > 0.05), except for a lower K concentration in both supplemented groups compared to the control with in addition a lower K concentration in the NaCl group compared with the TPC group. The bone markers (CTX and OC) showed no difference between treatments.

### 3.4. Macroscopic Examination of the Urine

Macroscopic evaluation of urinary samples taken in the barn showed no abnormalities, except for some grit and cloudy urine in the TPC group (8.3%) (Table 5).

The percentage of pigs in the slaughterhouse with grit and cloudy urine was also higher in the TPC group compared with the two other groups (*p* < 0.05). Overall, the percentage of pigs with stones and white flakes was lower than 3%.

### 3.5. Microscopic Examination of the Urine

The main crystals found in the urine taken at the farm and at the slaughterhouse were struvite, COD, calcite and amorphous crystals (Table 6).

These crystal types were also found at the slaughterhouse, but the prevalence was generally lower, especially for calcite. Calcite was the most prevalent crystal in urine taken at the farm (64% to 93%), whereas COD crystals were most prevalent in urine collected at the slaughterhouse (31% to 51%). The abundance of calcite crystals was highest in the citrate group (control 1.16; TPC 2.58; NaCl 1.26). Calcium oxalate monohydrate (COM) crystals were not found in any of the samples.

At the farm, the prevalence of amorphous crystals was significantly lower in the TPC group (4%) compared to the control (33%) and the NaCl (32%) group (*p* < 0.05). No significant differences were found between the groups at the farm for the other types of crystals.

At the slaughterhouse, significant differences in the prevalence of crystals were found between the groups for calcite (highest in TPC group) (*p* < 0.001), COD (highest in NaCl group) (*p* < 0.001) and struvite (lowest in NaCl group) (*p* < 0.05). The prevalence of amorphous crystals in urine collected at the slaughterhouse was 5% or less, and not significantly different between the groups.

### 3.6. Urinalysis

At the farm level, the urinary K concentrations and the urinary pH were highest in the TPC group (*p* < 0.001; Table 7).

The other parameters in the urine collected at the farm were not significantly different between the groups. At the slaughterhouse level, both supplementations had reduced urinary P and Mg concentrations.

There were large differences in the results between the two sampling sites. Most minerals had a lower concentration in the slaughterhouse samples with the exception of P (higher value) and magnesium (comparable values). The pH of the urine taken at the slaughterhouse was significantly lower than the pH of the samples taken at the farm.

## 4. Discussion

The present study did not show a beneficial effect of feed supplementation with TPC or NaCl on the prevalence of COD crystals in the urine of the fattening pigs. In addition, no significant influence was found on bone mineralisation parameters in serum or on the performance of the fattening pigs. Interestingly, there were substantial differences in the results of microscopic examination of urine and biochemical urinalysis parameters in samples taken at the farm versus samples taken at the slaughterhouse.

There were no significant differences in performance between the three groups. The overall performance results were within normal ranges, including the water-to-feed ratio. Recommended water-to-feed ratios in swine were reported to be between 2.0 and 2.5 [23,24,25]. The water consumption was very similar between the groups, whereas a higher intake was expected in the NaCl group. Higher NaCl intake increases the extracellular fluid osmolality, which directly stimulates the thirst centre and triggers antidiuretic hormone production [26]. The enhanced water intake and diminished urinary water loss can lead to a normalisation of plasma osmolality and plasma sodium levels. Upon the establishment of this new steady state, water intake might return to normal levels [12]. Additionally, Kitada et al. reported that higher NaCl intake in mice led to higher NaCl excretion without an increase in water excretion [27]. This phenomenon might explain why the water intake in the present study was not higher in the NaCl group.

Since reabsorption of calcium parallels sodium reabsorption in the proximal tubule and the loop of Henle, NaCl supplementation can result in NaCl-induced calciuria which can have a negative effect on bone tissue [13,28]. The bone markers and urinary excretion did not provide proof of this principle in the present study, maybe because the dosage was not high enough to force such changes, or because other actions were counteracting, leading to a net absence of an effect.

Similar to our previous study [3], only a few macroscopic findings were detected in the urine. Grit and cloudy urine were found in 7 to 8% of the samples taken at the farm and in the slaughterhouse. Stones or white flakes were detected in less than 3% of the samples collected in the slaughterhouse. They were not found in the samples collected at the farm, possibly because they are not easily expelled during spontaneous micturition.

Microscopic evaluation of the urinary sediment of the samples taken at the farm and the slaughterhouse showed that calcite, COD, struvite and amorphous crystals were the most prevalent. The prevalence was generally lower in the samples collected in the slaughterhouse. The decrease was most pronounced for calcite, likely due to the lower pH of the urine in the slaughterhouse samples that led to the resolution of those crystals.

The COD prevalence in this farm was slightly lower than the COD prevalence (52%) found in a previous study on the same farm [3], approximately two years before the present study. In the former study, samples were taken in the slaughterhouse, and calcite, struvite and amorphous were found in less than 10% of the samples. The high COD prevalence was the reason to test the TPC and NaCl supplementation as possible control measures in the present study. Unfortunately, no beneficial effects were obtained. This implies that the applied dosages are not effective under the present farm conditions to control or prevent COD crystals.

The high urinary potassium, citrate and pH in the TPC group were likely due to the alkalinisation property of TPC. Orally ingested TPC is mainly metabolised in the liver to potassium and bicarbonate, resulting in an alkaline load. Less than 10% of TPC escapes liver metabolism and is directly excreted in the urine [8,9]. Urinary sodium levels were highest in the NaCl group. The higher urinary Na levels can be considered as a reflection of the higher Na intake in this group in the steady state situation [29]. The absence of significant differences in specific gravity, an estimate of the concentration of the urine supernatant, between the NaCl group and the other groups indicates that the applied NaCl supplementation level (0.20%) was not sufficient to dilute the urine to levels that were detectable. The animals in the NaCl group were therefore able to cope with the increased NaCl intake without notable impact on their metabolism. Differences in urinary specific gravity might influence the formation of crystals. In the case of a low urine volume, increased crystalloid concentration can act as a risk factor for calcium stone formation [30].

Although it was not statistically significant (*p* = 0.088), it was unexpected that the TPC group had the highest calcium levels, expressed as standardised to creatinine levels, in the urine collected at the farm (6.16 versus 1.8 in the control group and 2.36 mmol/g creatinine in the NaCl group). The creatinine levels in the urine collected at the slaughterhouse were slightly lower (control), higher (TPC) or very similar (NaCl) compared to the levels in the urine collected at the farm. Studies in humans showed decreased calciuria when taking TPC or drinking bicarbonate-rich mineral water [31]. Orally administered citrate generates bicarbonate as it is mainly metabolised in the liver. The consequent rise in plasma bicarbonate concentration results in decreased bicarbonate resorption in the proximal tubuli and increased bicarbonate secretion by the intercalated cells type B in the collecting ducts. These processes lead to a rise in urinary pH, which was also the case in the present study, as the pH was significantly higher in the TPC group. To maintain electroneutrality in the kidney tubuli, Na, K or Ca must accompany the bicarbonate. In contrast to the good solubility of Na and K salts, calcium bicarbonate is poorly soluble and precipitates as calcium carbonate. This should normally result in lower soluble levels of Ca in the urine.

There were large differences between the biochemical results of urine samples collected during spontaneous voiding at the farm and urine samples collected in the slaughterhouse. The transport of the animals and the slaughter process induce stress in the animals, leading to metabolic acidosis and a decrease in the urinary pH [32,33]. The lower urinary pH in the slaughterhouse samples explains the lower urinary citrate concentrations compared to the spontaneously voided samples. The lower urinary pH is a result of glycolysis with lactic acid as the end product. To process the products of the glycolysis in the Krebs cycle, it is beneficial that citrate enters the mitochondria and is not lost in the urine.

In the slaughterhouse samples, P concentrations were higher, while Na, Cl and K concentrations were lower than in the samples taken on the farm. The higher P concentrations may be related to the lower pH of the urine, as in addition to ammonium, phosphate may act as a urinary buffer to eliminate acids. Mota-Rojas et al. found an increased haematocrit in pigs upon arrival at the slaughterhouse, due to splenic contraction and partly also to dehydration [34]. This state of relative volume depletion will activate the renin-angiotensin-aldosterone system and enhance Na and Cl reabsorption. Epinephrine released during stress situations drives K into the cells, leading to lower concentrations in serum and urine [26]. The lower levels of Ca in urine collected in the slaughterhouse can be explained by the fact that secretion and reabsorption processes for Ca and Na are similar in the kidney [35]. The large differences in results of microscopic examination of urine sediment and biochemical analysis of urine collected at the farm and the slaughterhouse indicate that results from the slaughterhouse might not fully reflect the situation on the farm, and, therefore, one should be very careful with the interpretation of results from samples collected at the slaughterhouse. In the case of microscopic examination, one should be aware that at least part of the crystals formed in alkaline urine can disappear because of the drop in urinary pH in the slaughterhouse.

The present study was based on a large number of animals. In total, 2318 pigs, distributed in nine successive batches of approximately 255–260 fattening pigs each, were included. The study spanned a period of 14 months, meaning that batches within each treatment group were raised in different periods of the year. Additionally, many different parameters were examined in the urine and the blood of the animals and different performance parameters were included as they are important for the farmer from an economic viewpoint. As the study was conducted on only one farm, with specific farm and management characteristics and urolithiasis problems, the results, however, cannot be generalised as such to other farms.

## 5. Conclusions

There were no beneficial effects of feed supplementation with TPC or NaCl on the prevalence of COD or other crystals in the urine of the fattening pigs at this specific farm having a history of a high occurrence of COD crystals. On the contrary, TPC has a clear promoting influence on the development of alkaline calcite crystals. The supplementation did not significantly influence bone mineralisation parameters or the performance of the pigs. Further research is warranted to identify effective control measures against urolithiasis. Interestingly, substantial differences were found between microscopic examination and biochemical urinalysis results from samples taken at the farm versus samples taken at the slaughterhouse, indicating that results from slaughterhouse samples might not fully reflect the situation on the farm.

## Figures and Tables

**Table 1 vetsci-09-00614-t001:** Number of animals and month of placement of the three treatment groups in the three compartments. There were three batches (replicates) of each treatment, and each batch of treatments was housed in a different compartment. The control group (control) received the normal feed used on the farm, and the tripotassium citrate (TPC) and the sodium chloride (NaCl) groups received the normal feed that was supplemented with 0.25% TPC and 0.20% NaCl, respectively.

Compartment	Treatment Group		
	Control	TPC	NaCl
1	n = 255 October 2019	n = 259 June 2019	n = 258 February 2020
2	n = 260 January 2020	n = 258 May 2020	n = 258 August 2019
3	n = 257 July 2019	n = 256 November 2019	n = 257 March 2020

**Table 2 vetsci-09-00614-t002:** Mineral composition ^a^ of the feed provided during the last weeks of the fattening period in each treatment group: the control group (control) received the normal feed used on the farm, the tripotassium citrate (TPC) and the sodium chloride (NaCl) groups received the normal feed supplemented with 0.25% TPC and 0.20% NaCl, respectively.

Parameter	Unit	Treatment Group						SEM	*p*-Value
		Control (n = 3)		TPC (n = 3)		NaCl (n = 3)			
		Mean	Median	Mean	Median	Mean	Median		
Ca	mg/kg	8647	8410	8140	7350	7160	7190	1404.71	0.430
Cu		13	12	15	16	20	24	6.52	0.670
Fe		231	225	270	246	299	296	76.75	0.491
K		6363	6380	6777	6730	6207	6310	259.04	0.099
Mg		1760	1650	1690	1580	1830	1610	334.86	0.670
Mn		83	82	82	79	78	76	6.57	0.561
Na		2210	2190	1930	1830	2930	2870	403.73	0.099
P		3983	4050	3720	3650	4003	4040	177.73	0.193
S		563	531	582	510	615	549	153.57	0.733
Zn		97	95	82	82	86	86	5.77	0.099
Cl		4040	4200	4383	4530	5327	5210	482.86	0.061
Na+K-Cl	meq/kg	145	142	135	135	136	123	15.84	0.561
dEB ^b^		218	77	203	68	191	64	14.78	0.670
Ca/P ratio		2.17	2.19	2.19	2.03	1.79	1.78	0.39	0.288

^a^ average and median values of the three replicates from each feed; SEM: standard error of the mean; ^b^ dEB: dietary electrolyte balance: Na+K-Cl+Ca+Mg-P-S.

**Table 3 vetsci-09-00614-t003:** Performance data ^a^ of fattening pigs in three treatment groups (mean and median of three batches in each group): the control group (control) received the normal feed used on the farm, the tripotassium citrate (TPC) and the sodium chloride (NaCl) groups received the normal feed supplemented with 0.25% TPC and 0.20% NaCl, respectively.

Parameter	Treatment						SEM	*p*-Value
	Control		TPC		NaCl			
	Mean	Median	Mean	Median	Mean	Median		
Duration (d)	114		115		117		3.95	0.670
Initial weight (kg)	34.4	34.4	33.8	34.0	31.3	31.9	4.07	0.463
Final weight (kg)	117.8	115.1	114.5	114.8	111.0	111.0	7.26	0.925
Average daily gain (g/d)	720	717.3	697	719	685	676	59.04	0.957
Average daily feed intake (kg/d)	2.18	2.21	2.07	2.04	2.11	2.13	0.21	0.733
Feed conversion ratio	3.01	3.05	2.94	2.83	3.09	3.20	0.24	0.733
Feed conversion ratio (25–115 kg)	2.62	2.78	2.58	2.59	2.78	2.80	0.29	0.491
Mortality (%)	2.84	3.47	2.20	1.17	2.72	2.71	1.41	0.670
Water consumption (L/d/pig)	4.41	4.52	4.44	4.65	4.44	4.41	0.19	0.995
Water: feed ratio	2.02	2.04	2.14	2.28	2.10	2.07		

^a^ average and median values of the three replicates from each group; SEM: standard error of the mean.

**Table 4 vetsci-09-00614-t004:** Serum concentrations (mean and median) of bone markers, parathyroid hormone (PTH) and minerals in fattening pigs of three treatment groups: the control group (control) received the normal feed used on the farm, the tripotassium citrate (TPC) and the sodium chloride (NaCl) groups received the normal feed supplemented with 0.25% TPC and 0.20% NaCl, respectively. Ten animals were sampled in each replicate at approximately 24 weeks of age (2 to 3 weeks prior to slaughter).

Parameter ^A^	Unit	Treatment						SEM	*p*-Value
		Control (n = 30)		TPC (n = 30)		NaCl (n = 30)			
		Mean	Median	Mean	Median	Mean	Median		
CTX	ng/mL	0.17	0.15	0.19	0.19	0.21	0.20	0.03	0.130
OC		73.60	72.85	79.16	84.25	74.19	75.65	4.78	0.431
PTH	ng/L	5.43	2.87	11.85	1.51	7.49	1.6	6.12	0.829
Ca	mmol/L	2.30	2.42	2.27	2.53	2.03	1.94	0.12	0.063
K		6.57 ^a^	6.17	5.10 ^b^	5.43	4.40 ^c^	4.22	0.48	<0.001
P		3.40	2.84	2.48	2.53	2.23	1.96	0.68	0.132
OC/CTX		433	360	416	353	353	384	94.05	0.556
OC x CTX		12.51	14.30	15.04	15.20	15.57	14.51	2.19	0.842

^A^ CTX: (bone resorption); OC: osteocalcin (bone formation); PTH: parathyroid hormone; Ca: calcium; K: potassium; P: phosphorus; SEM: standard error of the mean. ^a,b,c^ within a row, values with a different superscript differ significantly.

**Table 5 vetsci-09-00614-t005:** Macroscopic findings (stones/grit/white flakes) of the urine samples taken at the farm and in the slaughterhouse from fattening pigs given three dietary treatments: the control group (control) received the normal feed used on the farm, the tripotassium citrate (TPC) and the sodium chloride (NaCl) groups received the normal feed supplemented with 0.25% TPC and 0.20% NaCl, respectively.

	Treatment			*p*-Value
Farm	Control (n = 15)	TPC (n = 24)	NaCl (n = 31)	
Stones (%)	0	0	0	-
Grit, cloudy urine (%)	0 ^a^	8.3 ^a^	0 ^a^	-^d^
White flakes (%)	0	0	0	-
Slaughterhouse	Control (n = 126)	TPC (n = 144)	NaCl (n = 115)	
Stones (%)	2.6	2.6	0.8	0.486
Grit, cloudy urine (%)	0.9 ^a^	7.0 ^b^	2.3 ^a^	0.027
White flakes (%)	1.7	0.9	1.6	0.839

^a,b^ within a row, values with a different superscript differ significantly at *p* < 0.05 in the two-sided test of equality for column proportions. ^d^
*p*-values for pairwise comparisons were: Control–TPC *p* = 0.270; NaCl–TPC *p* = 0.100; Control–NaCl: no pairwise comparison possible.

**Table 6 vetsci-09-00614-t006:** Microscopic findings ^A^ of the urine samples taken at the farm and in the slaughterhouse from fattening pigs receiving three dietary treatments: the control group (control) received the normal feed used on the farm, the tripotassium citrate (TPC) and the sodium chloride (NaCl) groups received the normal feed supplemented with 0.25% TPC and 0.20% NaCl, respectively.

	Treatment			*p*-Value
Farm	Control (n = 15)	TPC (n = 24)	NaCl (n = 31)	
Calcite (%) (score ^B^)	93 (1.16)	91 (2.58)	64 (1.26)	0.092
COD (%)	53	64	43	0.373
COM (%)	0	0	0	-
Struvite (%)	33	13	42	0.113
Amorphous (%)	33 ^a^	4 ^b^	32 ^a^	0.025
Slaughterhouse	Control (n = 126)	TPC (n = 144)	NaCl (n = 115)	
Calcite (%)	1 ^a^	23 ^b^	3 ^a^	<0.001
COD (%)	31 ^a^	34 ^a^	51 ^b^	<0.001
COM (%)	0	0	0	-
Struvite (%)	21 ^a^	16 ^a^	7 ^b^	0.015
Amorphous (%)	4	4	5	0.775

^A^ Calcite: calcium carbonate; COD: calcium oxalate dihydrate; COM: calcium oxalate monohydrate; Struvite: magnesium ammonium phosphate. ^B^ The abundance of crystals was assessed using a scoring system ranging from one to five: score 1: 1–2 crystals, 2: 3–5 crystals, 3: 6–20 crystals, 4: 21–100 crystals; 5 > 100 crystals per microscopic field. ^a,b^ within a row, values with a different superscript differ significantly at *p* ≤0.05 in the two-sided test of equality for column proportions.

**Table 7 vetsci-09-00614-t007:** Results of urinalysis ^A^ of samples taken at the farm and in the slaughterhouse from fattening pigs receiving three dietary treatments: the control group (control) received the normal feed used on the farm, the tripotassium citrate (TPC) and the sodium chloride (NaCl) groups received the normal feed supplemented with 0.25% TPC and 0.20% NaCl, respectively.

		Treatment						SEM	*p*-Value
Farm	Unit	Control (n = 15)		TPC (n = 24)		NaCl (n = 31)			
		Mean	Median	Mean	Median	Mean	Median		
Ca	mmol/g creatinine	1.8	1.7	6.16	3.11	2.36	1.32	2.59	0.088
P		1.19	0.41	1.11	0.59	1.38	1.10	0.74	0.883
Mg		3.19	2.61	3.84	4.56	3.23	2.97	0.98	0.547
Na		18.01	13.14	35.51	37.26	36.55	30.93	12.21	0.182
Cl		29.21	22.62	33.68	31.47	39.69	43.72	9.53	0.055
K		35 ^a,b^	33	47 ^b^	51	18 ^a^	16	8.96	<0.001
Citrate		0.18	0.16	0.64	0.43	0.62	0.58	0.28	0.131
pH		7.46 ^a^	7.50	8.06 ^b^	8.50	7.11 ^a^	7.00	0.27	0.001
Specific gravity		1.024	1.020	1.027	1.028	1.022	1.030	0.004	0.748
Slaughterhouse		Control (n = 50)		TPC (n = 29)		NaCl (n = 30)			
Ca	mmol/g creatinine	0.58	0.48	0.60	0.57	0.58	0.47	0.13	0.968
P		7.80 ^a^	6.52	4.60 ^b^	4.88	5.54 ^b^	4.78	1.09	<0.001
Mg		3.17 ^a^	3.17	2.74 ^a,b^	2.72	2.36 ^b^	2.02	0.38	0.013
Na		15.3	12.3	13.1	12.2	16.3	15.6	3.54	0.551
Cl		8.7	8.04	9.1	7.75	7.4	9.32	1.65	0.483
K		13.2	11.24	12.8	13.2	10	12.7	2.05	0.117
Citrate		0.13	0.09	0.12	0.08	0.15	0.13	0.04	0.772
pH		5.97 ^a^	6.0	6.46 ^b^	6.5	6.28 ^b^	6.5	0.12	<0.001
Specific gravity		1.023	1.024	1.024	1.024	1.023	1.023	0.001	0.366

^A^ the values of Ca, P, Mg, Na, Cl, K and citrate are expressed as standardised to creatinine levels and consequently, expressed as (mmol/Lit)/(gram creatinine/Lit). The creatinine levels at the farm and the slaughterhouse were as follows: Control: 4.18 and 3.87; TPC: 3.07 and 3.87; NaCl 3.09 and 3.12. Results are average and median values of the three replicates from each group; SEM: standard error of the mean. ^a,b^ within a row, values with a different superscript differ significantly at *p* ≤0.05 in the two-sided test of equality for column means. Tests do not assume equal variances and are corrected for all pairwise comparisons with the use of the Tamhane T2 post hoc test.

## Data Availability

Data are contained within the article and Appendix A.

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
