# Peer review of "Effect of Feed Supplementation with Tripotassium Citrate or Sodium Chloride on the Development of Urinary Calcium Oxalate Crystals in Fattening Pigs"

_vetsci, 2022, doi:10.3390/vetsci9110614_

Round 1
Reviewer 1 Report
This paper presents an interesting study of the possibility of influencing the incidence of urolithiasis in pigs by dietary changes. I have a few questions about the paper and would ask for some information to be added.
line 35: You wrote: "The pathogenesis of urolithiasis is not yet fully elucidated"
Which urolithiasis? Calcium oxalate? In pigs or in general? Please specify.
Chapter 2.3.3.: Please specify the time between urine sampling and examination. And then consider the influence of storage conditions on the presence of various types of crystals.
Please do the same for chapter 2.3.4.
Lines 182-185
Were all samples examined using both the dipstick and the refractometer?
Chapter 3 - Results
Why are the non-normally distributed data expressed as mean instead of median and range?
Is it possible to add any correlation, at least between the specific gravity and the presence of crystals in urine?
Please check:
line 83: feed 2 (40-55 kg), feed 2 (55-70 kg)
line 441: Al blood
Author Response
The authors would like to thank the reviewers and the editor for the careful evaluation of the manuscript and the constructive comments. We have revised the manuscript accordingly.
Comments Reviewer 1:
line 35: You wrote: "The pathogenesis of urolithiasis is not yet fully elucidated" Which urolithiasis? Calcium oxalate? In pigs or in general? Please specify.
Answer:
Especially the pathogenesis of urolithiasis in pigs is not yet fully elucidated. In contrast to other animals such as dogs and cats, literature concerning the condition in pigs is scarce. See line 26, 27 and 35 for the revisions in the text.
Chapter 2.3.3.: Please specify the time between urine sampling and examination. And then consider the influence of storage conditions on the presence of various types of crystals.
Answer:
This is a good comment since time between sampling and examination, and an increased storage time can influence crystal formation. The information has been added to the text (See line 166-168). Also, two references have been added (ref 21 and 22)
Please do the same for chapter 2.3.4.
Answer:
This has been done. See line 191 where we refer to the previous prevalence study
Lines 182-185
Were all samples examined using both the dipstick and the refractometer?
Answer:
Yes, pH en SG examination was done on all the samples. This has been added to the text, see line 18.
Chapter 3 - Results
Why are the non-normally distributed data expressed as mean instead of median and range?
Answer:
Most of the data were normally distributed, a few did not follow a normal distribution. To accomplish for this and present the data in a consistent way, we have added the median values in addition to the mean value in table 2 (mineral feed composition), table 3 (performance data), table 4 (blood parameters) and table 7 (results of urinanalysis). In order not to make the tables too complex and because most of the data were normally distributed, but we did not add the range of the data in the tables.
Is it possible to add any correlation, at least between the specific gravity and the presence of crystals in urine?
Answer:
Yes, we have added the correlations between SG and possible crystal formation in the revised version of the paper. See Line 378 and line 382-384. Also, we have added an extra reference (number 30).
Please check: line 83: feed 2 (40-55 kg), feed 2 (55-70 kg)
Answer:
Indeed good comment. We have changed feed 2 to feed 3 (line 80).
line 441: Al blood.
Answer:
Thanks for noticing this typing error. We have changed it in “all blood”. See line 448.
Reviewer 2 Report
The article is quite original and novel and has also been written clearly. It makes a significant contribution to the body of knowledge in swine nutrition, there are several minor issues that need to be taken into account before the manuscript is considered for publication in this journal.
L83 "feed 1 (25-40 kg), feed 2 (40-55 kg), feed 2 (55-70 kg), feed 4 (70-95 kg)" one of them should be feed3
please go through the manuscript again to revise the similar errors.
Author Response
The authors would like to thank the reviewers and the editor for the careful evaluation of the manuscript and the constructive comments. We have revised the manuscript accordingly.
Comments Reviewer 2:
The article is quite original and novel and has also been written clearly. It makes a significant contribution to the body of knowledge in swine nutrition, there are several minor issues that need to be taken into account before the manuscript is considered for publication in this journal.
L83 "feed 1 (25-40 kg), feed 2 (40-55 kg), feed 2 (55-70 kg), feed 4 (70-95 kg)" one of them should be feed3
Answer:
Indeed good comment. We have changed feed 2 to feed 3 (line 80).
Please go through the manuscript again to revise the similar errors
Answer:
We have checked the entire manuscript and made some additional minor revisions.